# Hyperspectral Data Compression Using Fully Convolutional Autoencoder

**Riccardo La Grassa** [1,*,†] , **Cristina Re** [1,†] , **Gabriele Cremonese** [1,†] and **Ignazio Gallo** [2,†]

1   National Institute for Astrophysics (INAF), 35100 Padua, Italy; cristina.re@inaf.it (C.R.); gabriele.cremonese@inaf.it (G.C.)
2   Department of Theoretical and Applied Science, University of Insubria, 21100 Varese, Italy; ignazio.gallo@uninsubria.it
*   Correspondence: riccardo.lagrassa@inaf.it
†   These authors contributed equally to this work.

**Abstract:** In space science and satellite imagery, better resolution of the data information obtained makes images clearer and interpretation more accurate. However, the huge data volume gained by the complex on-board satellite instruments becomes a problem that needs to be managed carefully. To reduce the data volume to be stored and transmitted on-ground, the signals received should be compressed, allowing a good original source representation in the reconstruction step. Image compression covers a key role in space science and satellite imagery and, recently, deep learning models have achieved remarkable results in computer vision. In this paper, we propose a spectral signals compressor network based on deep convolutional autoencoder (SSCNet) and we conduct experiments over multi/hyperspectral and RGB datasets reporting improvements over all baselines used as benchmarks and than the JPEG family algorithm. Experimental results demonstrate the effectiveness in the compression ratio and spectral signal reconstruction and the robustness with a data type greater than 8 bits, clearly exhibiting better results using the PSNR, SSIM, and MS-SSIM evaluation criteria.

**Keywords:** autoencoder; data compression; remote sensing; satellite images

## 1. Introduction

Currently, deep learning models are successfully applied in any area of research topic achieving remarkable results in classification [1], segmentation [2], super-resolution [3,4] and in different environments such as biology [5], medical imaging [6], geology [7], remote sensing [8], and space science [9]. In the latter, deep learning can be a powerful instrument able to resolve complex tasks such as self-driving [10], system recommendation [11], data compression [12–14], and 3D surface reconstruction [15]. Data compression is the main focus that we will discuss here considering the implications for space science and Earth observation [8,16]. It is a necessary step for the space mission context, and it can be useful to the limited hardware mainly, to preserve as much as possible the hardware life on board (e.g., satellite, drone, rover) and process the huge incoming information. The deep learning models (e.g., autoencoder) can be used for signal classification, segmentation, data compression, clustering discovery, and signal generation. Traditional image compression algorithms such as JPEG [17] and JPEG2000 [18] act on cosine and wavelet transform jointly with a quantization step and entropy approach to compress the image. Other methods as CCSDS 123.0 are based on discrete wavelet transform (DWT), which uses low amounts of memory and computational resources and was developed to balance between compression performance and complexity with a particular focus on space science context. Compared to the JPEG2000 algorithm, it achieves lower performance in terms of image reconstruction [19]. The hand-crafted codec step is not optimal and flexible for all types of signals source (e.g., hyperspectral data) and offers low performance than the deep learning models in compression ratio and signal reconstruction tasks. Even though JPEG families algorithms are still

used due to simplicity, it causes the generation of artifacts, perceptual distortions, and blur, above all at lower bit-rate ranges, and for this reason, many postprocessing techniques have been applied. Deep learning models can play a crucial role in satellite imagery both in real-time and postprocess steps. In real-time because today, it is possible to handle the huge computation of parallel paradigms (e.g., Pytorch, Tensorflow) directly on-board using dedicated hardware [20] or in postprocess exploiting the workstation computational power. Especially, the autoencoder model can be applied in dimensionality reduction, clustering discovery, and classifications or can be used as generative learning models. In [21], they proposed a wavelet-based deep autoencoder network based on image compression using Haar transformation preprocessing to get different frequency components in an image and they used convolutional operators in encoder/decoder to reconstruct the original source given the latent space obtained from the encoder step. In [22], a different architecture is proposed but using the same Wavelet frequency decomposition, showing better results than the classical algorithms (e.g., JPEG, JPEG2000). Although the leadership of the deep models was proved by all the experimental results in the literature, both methods use only dataset composed by 3-channels without investigating over spectral data information. In [23], the authors proposed a feedforward neural network as an end-to-end model to reproduce its input by learning the identity function. The model is trained considering many selected bands (e.g., channels) conducting extensive experiments showing good robustness to noise and analyzing three different loss functions. Due to the fully connected layers of the architecture proposed by [23,24], as the spectral image size increases, the number of model parameters becomes high, slowing the training time process and cutting off the model flexibility as the dataset changes, and in addition, the last fully connected layers can introduce noise and destabilize the learning capability. Moreover, it is mandatory to use a batch-size iterator to avoid memory saturation (e.g., out of memory), and all models with a high number of parameters may encounter this problem. Autoencoders confirm the capability to get better features coded from signals source than existing algorithm as JPEG and JPEG2000, remaining competitive in performance with the latter using the pretrained models [21,25,26]. Furthermore, the main model can generalize the reconstruction source independently by the data type (e.g., >8 bits per pixel) or format type data (e.g., eyes fish camera images).

We know the reconstruction models (e.g., autoencoder-based) work well from 1 to 3 channels as widely supported by the literature [12,14,27]. Thus, the following question arises naturally:

*Is it still true over multispectral and hyperspectral data information?*

*Given a training data set, has the model the capability to generalize the learning process using 16 bits per pixel band of the unseen spectral source input instead of 8 bits?*

*What kind of data compression ratio can be achieved by keeping the reconstructed output signal as high as possible? Can we achieve better performance than classical image compressors such as JPEG and JPEG2000?*

To complete the lack of tests and analysis of the autoencoder performances on spectral sources and to avoid the classic feedforward due to the above-mentioned problems, in this work, we build an autoencoder for the compression of spectral signals based on convolutional linear operators and we demonstrate through experimental results that the proposal surpasses the classic JPEG and JPEG2000 compression algorithms and neural networks used as benchmarks, in terms of peak signal–noise ratio (PSNR) and structural similarity index (SSIM) and about data compression.

The scientific contribution of this work can be summarized as follows:

- Development of a spectral signals compressor based on deep convolutional autoencoder (SSCNet), analysing its learning process and evaluating it in terms of compression and spectral signal reconstruction over spectral datasets and Imagenet-ILSVRC2012 benchmark.

- Definition of two datasets come from the ESA repository (Lombardia Sentinel-2 satellite imagery and VIRTIS-Rosetta hyperspectral data) and development of a python parser useful to read and handle the calibrated data images.
- Release the PyTorch code for SSCNet, the pretrained models and the parser software available in [28].

The paper is organized as follow: In Section 2, we describe in detail the proposed network for spectral signal compression and the experimental setup. In Section 3, the description of the datasets, discussion results in terms of compression ratio, image reconstruction analysis and qualitative spectral signals visualization are presented. In the last part of this paper, we present the Section 4.

## 2. Spectral Signals Compressor Network

The overview of the proposed spectral signal compressor based on convolutional autoencoder is shown in Figure 1. In the encoder block, we scale $3\times$ the spectral source by using max-pooling operation followed by a linear layer to set the final latent space size (data compression output). The decoder block mirrors the encoder flow using the convolution transposed with the relative kernel and stride setup. The architecture shown in Figure 1, is correlated to all experiments carried out over the Lombardia Sentinel-2 dataset (subtle differences are present in the model used in other datasets considered; see the repository for details [28]). From now, to simplify the nomenclature of the network, we refer to it as SSCNet. As preprocessing, we apply a normalization on the spectral data reporting all data in a range between $[0,1]$ and fed into the end-to-end model. The last operator of SSCNet is a sigmoid activation function that maps the output of the decoder block into the $[0,1]$ range. More formally, given a spectral cube image $x$ in the format:

$$x \in \mathbb{R}^{B \times H \times W}$$

where $B = Bands$, $H = Line$, $W = Samples$, we search a mapping function as $y = f_0(x)$ and a function useful to approximate the identity function as $\tilde{x} = g_\theta(y)$ where $x$, $\tilde{x}$, and $y$ are the spectral signals source, the approximation identity signal reconstructed, and the latent space signals (data compression size), respectively. To obtain an arbitrary data compression, we use a dense linear layer into the last part of the encoder block such that it is possible to analyze spectral signals compressed and the relative spectral cube reconstructed thanks to the decoder mirrored block. We apply downsampling using max-pooling operators and deconvolutions linear operator as upsampling into the decoder step. To obtain an approximation of the spectral signals source, we use the binary cross-entropy (BCE Loss) that builds the error surface between the target and the input distributions. More formally, it is described as:

$$
\begin{aligned}
l(x,y) &= L = \{l_1, ..., l_N\}^T, \\
l_n &= -w_n[y_n \dot{l}og(x_n) + (1-y_n)\dot{l}og(1-x_n)]
\end{aligned}
\tag{1}
$$

where $N$ is the batch size and $x$, $y$ are input and target, respectively. If $x_n$ is 0 or 1, the log will be set to $-\infty$, and the BCE Loss clamps its log function output to be greater than or equal to a fixed value in order to have a finite loss (rather than infinite value) as well-described into the PyTorch documentation [29]. For the above-mentioned reasons and the nature of the BCE Loss, we apply a sigmoid function into the last part of the decoder block. We chose PReLU (parametric rectified linear activation function) instead of ReLU for the $a$ learnable parameters of the elementwise function:

$$PReLU(x) = max(0,x) + a \cdot min(0,x) \tag{2}$$

In our case, we set $a$ equal to the number of the channels (bands) given in input from the previous layer SSCNet step (except into the first part of the decoder, where we have learnable parameters for each element of the flattened data information). We analyzed

and discussed the difference comparison using ReLU and PReLU activation functions in Section 3.

#### Encoder

```
Conv2d(c, 64, k=3, p=1)
PReLU(64)
Conv2d(64, 128, k=3, p=1)
PReLU(128)
MaxPool2d(2, 2),
Conv2d(128, 256, k=3, p=1)
PReLU(256)
MaxPool2d(2, 2)
Conv2d(256, 256, k=3, p=1)
PReLU(256)
MaxPool2d(2, 2)
Flatten()
Linear(256*H*W, latent space)
```

#### Decoder

```
Linear(latent space,256*H*W)
PReLU(256*H*W)
Data Reshape(batch size, 256 , H , W)
ConvTranspose2d(256, 256, k=2, s=2)
PReLU(256)
ConvTranspose2d(256, 128, k=2, s=2)
PReLU(128)
ConvTranspose2d(128, 64, k=2, s=2)
PReLU(64)
ConvTranspose2d(64, c, k=3, s=1, p=1)
Sigmoid()
```

**Figure 1.** Overview of the architecture proposed for the spectral signals compression. c, k, p, and s are channels, kernel size, padding, and stride used, respectively. In the encoder block the stride is set up to 1. The latent space represents the data compression size of the last linear layer. This specific version model is applied over Lombardia Sentinel-2 dataset and little variations on the other dataset (see repository code [28] for details).

We used ADAM optimizers [30] able to find global minima or a good approximation of it from the error surface generated by the BCE loss function. We used a batch size of 128 and trained SSCNet for 400 epochs. The learning rate of $10^{-4}$ is handled by cosine annealing decay achieving 0 to the last epochs. This last technique is useful to stabilize the learning process step. The optimal convergence in terms of the image reconstructed is achieved much earlier than 400 epochs (we observe little improvements in terms of cents into the last 150 epochs over 400 considered). The model validation and metrics used to evaluate the effectiveness of the SSCNet will be discussed in the Section 3.

### 3. Experimental Results

#### 3.1. Datasets

To prove the effectiveness and robustness of the proposal, we built two types of datasets in remote sensing and space context, *Lombardia Sentinel-2 dataset* and *VIRTIS-Rosetta dataset*. The main motivation is related to the data type of these datasets. The first is a multispectral one composed by 9 bands with a spatial resolution of 10 and 20 m, and the second one is an hyperspectral dataset composed by 432 bands captured by VIRTIS instrument on-board the Rosetta space mission. We were also motivated to use these datasets for the reasons of the data type values; in fact, all data types are expressed in 16 bits (unsigned integer) for the Lombardia Sentinel-2 dataset and in float-32 for the VIRTIS-Rosetta dataset.

#### 3.1.1. Lombardia Sentinel-2 Dataset

We built a subset (100k instances) of multispectral signal information from the Sentinel2 satellite over the Lombardia region in Italy. This dataset has spatial resolutions of 10 and 20 m, both corresponding to the period of 2017. Each spectral cube has 9 bands (channels) from VNIR (visible and near-infrared). The train-test split to 80:20 ratio, respectively, each multispectral images is $9 \times 48 \times 48$ with unsigned integer 16 bits (range = $[0, 2^{16} - 1]$), and all data in the cube are given in the Tiff format.

#### 3.1.2. VIRTIS-Rosetta dataset

The VIRTIS (Visual IR Thermal Imaging Spectrometer) (see the instrument details in [31]) is part of the scientific payload of the Rosetta Orbiter, and it has detected and ana-

lyzed the evolution of specific signatures as the specifics bands of minerals and molecules until to ended with the landing of the Rosetta Orbiter on 30 September 2016. The identification of spectral features was the primary goal of the Rosetta mission and had allowed to identify the natures of the main constituent of the 67-P Churyumov-Gerasimenko. The complex instrument is composed by different parts and it covers a wide wavelength range from near UV to the near IR making it an essential instrument to analyze the global (e.g., albedo) and local properties (minerals signatures). The VIRTIS spectrometer has been successfully applied also in other planetary missions such as Venus Express and NASA-Dawn, producing a huge amount of high-quality data. More precisely, the spectral range (nm) covered a range from [220.1–1046] for visible bands to 5 micron using VIRTIS-H part data instrument. We focus on visible bands for this paper of the calibrated data (DATA) using the full bands available. In this work, we built a dataset of the calibrated data selected from the NASA repository with a tile of $64 \times 64$ unitary spectral cropping of the relative signal sources. In general, before feeding spectral data into the SSCNet, we develop a parser useful to read the VIRTIS data format [32] available from the NASA site [33], then, we apply a train/test split in 80:20 ratio necessary to train the SSCNet model and validate it. SSCNet model can cover a fundamental role in similar future missions to handle and process the huge amount of spectral data information coming from these kind of instruments. In fact, autoencoder models can reduce the access memory (writing and reading) from internal satellite storage due to compression data learned in the training step, and depending on the compression ratio applied, it can store a greater amount of data than the classical compressor (e.g., JPEG2000) algorithm can, maintaining a competitive spectral signal reconstruction (decoder part) from compressed information (encoder part).

### 3.1.3. Normalization

The normalization process is widely used in data preprocessing, guaranteeing fast convergence in learning terms. This process removes the difference in magnitude between features closing to zero (e.g., max/min normalization), which benefits learning [34]. Convergence is usually faster if the average of each input variable over training data is close to zero. In the case in which we have all positive input values, the weight updated will have the same sign of the scalar error computed. Therefore, if the weight vector must change direction, it can only do so by *zigzagging*, which is inefficient and can slow the learning process (see Section 4.5 of [35]). In Rosetta experiments, we apply a spectral normalization (e.g., max/min) for each spectral signal of the dataset in the data preprocessing step; then, we use these values for each instance of the training set assuring a correct scaling per band (see Algorithm 1). In this way, we can take advantage in terms of convergence according to the above-mentioned information. Per instance, max/min normalization (local normalization) is strongly not recommended in data type >8 bits, because we *change* the shape of the original statistical data distribution (e.g., check the kurtosis and skewness modified), above all, when we have many outliers in the dataset considered. In the experiments, we used a spectral normalization per band over the VIRTIS-Rosetta dataset in such a way to have a proper normalization per band (432 bands) and a min/max normalization over all bands (9 bands) in the Lombardia Sentinel-2 dataset.

### 3.2. Training Batch Strategy in High-Spatial Resolution Input

Considering the high spatial resolution and the spectral information on a large number of bands, we adopt a subdivision strategy on each multi/hypercube $x \in \mathbb{R}^{B \times H \times W}$ to reduce the RAM consumption of the GPU (graphic processing unit) and avoid the out of memory error (see Figure 2).

---

**Algorithm 1** Pseudo code of the training step. We apply a min/max normalization per channel taking all min/max from a preprocessing dataset step; then, we feed into SSCNet $x_j$, which returns the compressed data. Finally, we use the compressed data and feed it into the SSCNet decoder module for the image reconstruction. The total error will be given from the binary cross-entropy error between the decoded image reconstructed and the original source, and the error backpropagation is applied for the learning process.

---

**function** TRAINING STEP(*X, epochs*)
    Input: Dataset instances, epochs
    Output: Data compressed, Data reconstructed
    **for** $n \in range(epochs)$ **do**
        **for** $x_i \in X$ **do**
            $x_j$ = Min/Max per Channel Normalization($x_i$)
            encoded = SSCNet encoder($x_j$)
            decoded = SSCNet decoder(encoded)
            $l$ = bce loss(decoded, $x_j$)         ▷ Binary cross-entropy Equation (1)
            Error $l$ backpropagation()         ▷ Error Backpropagation
        **end for**
    **end for**
**end function**

---

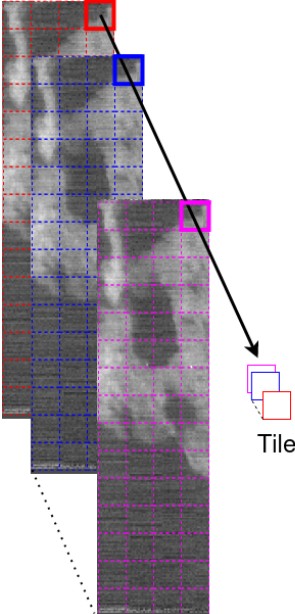

**Figure 2.** Subdivision strategy of multi/hypercube images.

The number of tiles $x_i \in X$ generated for each cube can be variable and depends on the available RAM of the GPU; in our case, we divide each multispectral cube of the Lombardia Sentinel-2 dataset into tiles of size $9 \times 48 \times 48$, and for the VIRTIS-Rosetta dataset, in tiles of size $432 \times 64 \times 64$; then, we select the hypercubes randomly to improve the learning capability. This training strategy is successfully applied in deep learning context on many remote sensing tasks [36].

### 3.3. Generalization Capability

To demonstrate the versatility and effectiveness of SSCNet over the RGB dataset, we conduct experiments on Imagenet. It is a huge dataset that contains a number greater than 1.2M of RGB images and 1000 different classes. As a test set, we used the Kodak Photo CD dataset as the benchmark. This last consists of 24 ($3 \times 768 \times 512$) images that contain landscapes, portraits, and humans. In this experiment, we demonstrate the generalization capability of a universal compressor trained on a huge dataset, over a never seen test set. In this experiments we used Adam optimizer with a learning rate of $10^{-4}$ and handled by cosine annealing decay achieving 0 to the last epochs (set up to 200). Since we use patch $3 \times 32 \times 32$ of the Kodak images, we run 20 times the test set over different random patch and average the metrics considered, for a fair comparison with the benchmarks used. Since

the evaluation process is not a trivial task for metric reasons, we do not report the peak signal-to-noise ratio (PSNR) only because it is biased toward algorithms that have been tuned to minimize L2 loss. We use Multi-Scale Structural Similarity (MS-SSIM) [37] and Structural Similarity (SSIM) [38], which are well-established and popular metrics for lossy compression algorithms comparison. For a fair comparison, we compare SSCNet with [25], which proposed a general architecture for compressing based on RNNs, content-based residual scaling, and a new variation of GRU where they achieved better performance than JPEG. Both works are comparable because we use a huge dataset also and validate the model over the same test set used as benchmark (Kodak dataset), furthermore, we used the same bits per pixel ratio (bpp) in different learning processes. We report all the results in Figures 3 and 4, showing an improvement across all baseline compared in terms of generalization reconstruction capability.

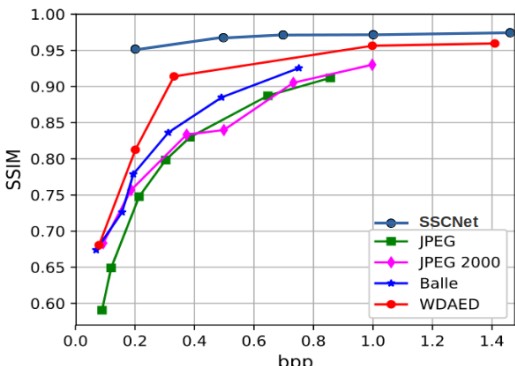

**Figure 3.** Rate distortion curve on the Kodak dataset using SSIM as evaluation metric over bits per pixel. We used all the results reported in [21,26] as benchmarks and demonstrate to overcome the reconstruction capability of the SSCNet than the others baseline (Balle and WDAED are based on convolutional neural networks) over a 3 channels dataset. We highlight our use of Imagenet-ILSVRC2012 as the training set and Kodak as the test set.

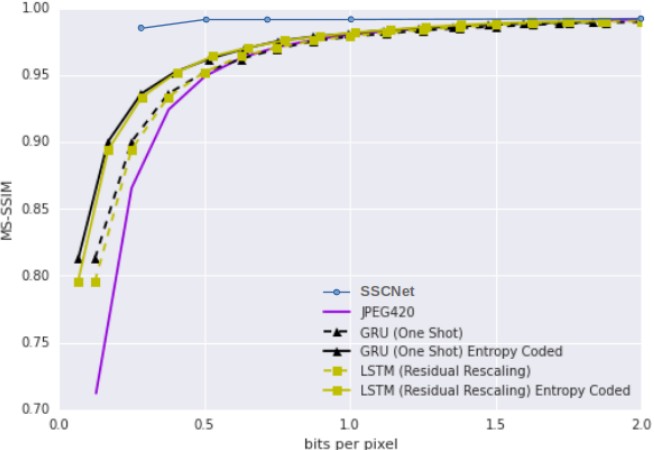

**Figure 4.** Rate distortion curve on the Kodak dataset using MS-SSIM as evaluation metric over bits per pixel. We show all the results reported by [25] demonstrate the reconstruction capability of the SSCNet using 3 channels dataset. We highlight our use of Imagenet-ILSVRC2012 as the training set and Kodak as the test set.

### 3.4. Coding and Spectral Signal Reconstruction Efficiency

In this subsection, we analyze the performance changing the compression ratio and evaluating the spectral images reconstruction achieved by the proposed method and comparing it with JPEG and JPEG2000. In addition, we analyzed the SSCNet behavior using PReLU and ReLU activation functions. To prove the effectiveness of the SSCNet

model, we evaluated the method over the test set built; then, we conducted a quantitative and qualitative analysis using the well-known metrics widely used in the literature (peak signal-to-noise ratio (PSNR), structural similarity (SSIM), and feature similarity index (FSIM)). In Table 1, we reported the impressive metric accuracies achieved by SSCNet with a compression ratio 20:1 over 400 epochs training (using ReLU) and in Table 2 using the same architecture with PReLU function. Although the performance is similar, we report the percentage bands variation (see Table 3), suggesting a slight improvement due to the learnable parameters in the PReLU notation formula; more formally, we define the variance percentage as:

$$\frac{PSNR_{PReLU} - PSNR_{ReLU}}{PSNR_{PReLU}} \cdot 100 \qquad (3)$$

**Table 1.** Compression ratio (20.7):1 over 400 epochs ReLU for the Lombardia Sentinel-2 dataset.

| Bands | PSNR | SSIM | FSIM |
|-------|------|------|------|
| 1 | 48.106 | 0.984 | 0.984 |
| 2 | 49.427 | 0.988 | 0.984 |
| 3 | 49.553 | 0.987 | 0.983 |
| 4 | 50.914 | 0.993 | 0.991 |
| 5 | 49.205 | 0.990 | 0.990 |
| 6 | 48.085 | 0.987 | 0.989 |
| 7 | 44.259 | 0.967 | 0.975 |
| 8 | 50.581 | 0.993 | 0.991 |
| 9 | 50.458 | 0.992 | 0.991 |

**Table 2.** Compression Ratio (20.7):1 over 400 epochs PReLU activation function for the Lombardia Sentinel-2 dataset.

| Bands | PSNR | SSIM | FSIM |
|-------|------|------|------|
| 1 | 48.440 | 0.985 | 0.985 |
| 2 | 49.709 | 0.988 | 0.985 |
| 3 | 49.783 | 0.988 | 0.984 |
| 4 | 51.318 | 0.993 | 0.991 |
| 5 | 49.569 | 0.991 | 0.991 |
| 6 | 48.542 | 0.988 | 0.990 |
| 7 | 44.527 | 0.969 | 0.977 |
| 8 | 51.043 | 0.993 | 0.992 |
| 9 | 50.876 | 0.992 | 0.991 |

**Table 3.** Percentage variation comparison over Lombardia Sentinel-2 dataset between PReLU and ReLU activation functions (gain percentage respect to PReLU).

| Bands | ΔPSNR | ΔSSIM |
|-------|-------|-------|
| 1 | 0.6895 | 0.10152 |
| 2 | 0.5673 | 0.0000 |
| 3 | 0.4620 | 0.10121 |
| 4 | 0.7872 | 0.0000 |
| 5 | 0.7343 | 0.1009 |
| 6 | 0.9414 | 0.1012 |
| 7 | 0.6018 | 0.2063 |
| 8 | 0.9051 | 0.0000 |
| 9 | 0.8216 | 0.0000 |

In Table 4, we report results in image reconstruction by SSCNet than the classic compressor algorithm as JPEG and JPEG2000. We obtain a greater spectral signal reconstruction over all bands (Lombardia sentinel dataset) than the other compressors in PSNR and SSIM metrics analyzed. JPEG2000 algorithm has been used for each band available (9 bands), showing a better compression ratio and reconstruction than the JPEG algorithm. Note that in the JPEG compressor test, we truncate the data from 16 bits to 8 bits due to the compression inefficient classical algorithm and apply it for single bands.

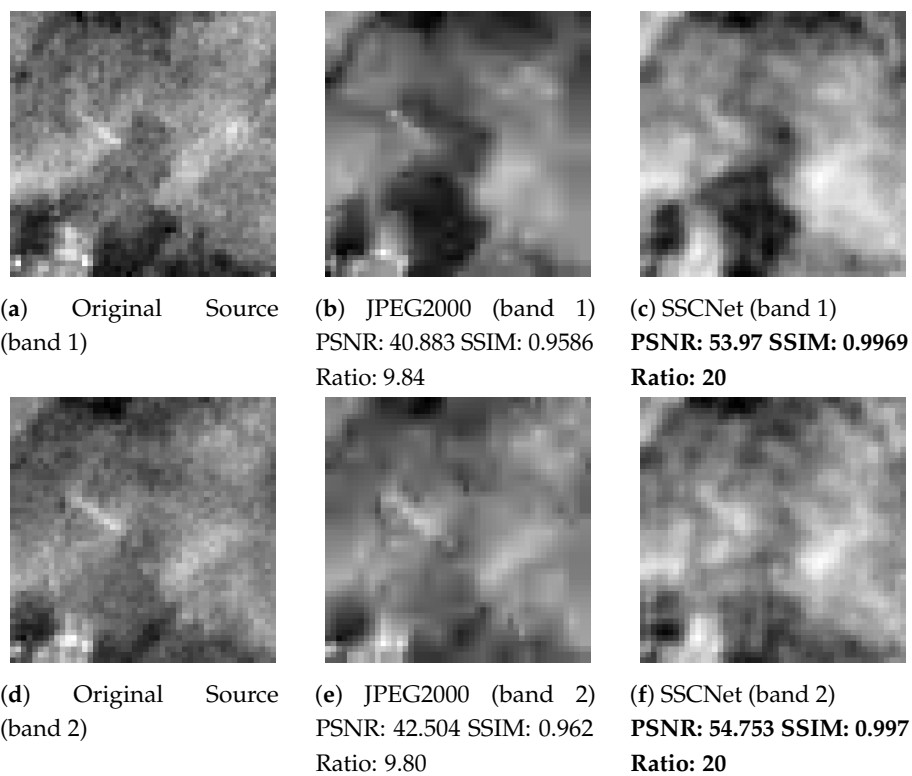

(**a**) Original Source (band 1)

(**b**) JPEG2000 (band 1) PSNR: 40.883 SSIM: 0.9586 Ratio: 9.84

(**c**) SSCNet (band 1) **PSNR: 53.97 SSIM: 0.9969 Ratio: 20**

(**d**) Original Source (band 2)

(**e**) JPEG2000 (band 2) PSNR: 42.504 SSIM: 0.962 Ratio: 9.80

(**f**) SSCNet (band 2) **PSNR: 54.753 SSIM: 0.997 Ratio: 20**

**Figure 5.** Comparison using the compression ratio 20:1 for SSCNet and (9.7):1 for JPEG2000 with a data type 16 bits on the original source tile (**a**,**d**) using JPEG2000 algorithm applied over all bands in (**b**,**e**) and SSCNet signal reconstructed in (**c**,**f**). We show only bands 1 and 2.

**Table 4.** Benchmark comparison between two classical compressor algorithms (JPEG, JPEG2000) and the SSCNet AutoEncoder based over the Lombardia sentinel dataset.

| Bands | JPEG Ratio (2.5):1 | | JPEG2000 Ratio (9.7):1 | | SSCNet Ratio 20:1 | |
| --- | --- | --- | --- | --- | --- | --- |
| | PSNR | SSIM | PSNR | SSIM | PSNR | SSIM |
| 1 | 19.171 | 0.908 | 35.317 | 0.925 | 48.440 | 0.985 |
| 2 | 19.302 | 0.906 | 36.249 | 0.923 | 49.709 | 0.988 |
| 3 | 19.404 | 0.899 | 35.710 | 0.912 | 49.783 | 0.988 |
| 4 | 21.722 | 0.945 | 36.219 | 0.929 | 51.318 | 0.993 |
| 5 | 21.561 | 0.948 | 36.570 | 0.932 | 49.569 | 0.991 |
| 6 | 21.523 | 0.949 | 36.354 | 0.931 | 48.542 | 0.988 |
| 7 | 18.798 | 0.916 | 35.767 | 0.929 | 44.527 | 0.969 |
| 8 | 21.866 | 0.943 | 37.444 | 0.946 | 51.043 | 0.993 |
| 9 | 21.904 | 0.942 | 35.847 | 0.938 | 50.876 | 0.992 |

**Table 5.** Compression ratio achieved by SSCNet using different latent space output (ls) or with the last convolution version. We report the PSNR between the original spectral source and the output of SSCNet (PSNR is averaged over all bands of the VIRTIS-Rosetta dataset). We report the results of GNN model based-on VGG16.

| Compression Ratio SSCNet | avg PSNR |
|---|---|
| 7:1 ($ls = 1024 \times 16 \times 16 \times 32$) last conv | **<u>67.677</u>** |
| 27:1 ($ls = 1024 \times 8 \times 8 \times 32$) last conv | 66.79 |
| 177:1 ($ls = 10,000 \times 32$) | **64.845** |
| 353:1 ($ls = 5000 \times 32$) | 64.841 |
| 1769:1 ($ls = 1000 \times 32$) | 64.729 |
| Ref. [24] GNN model (VGG16 + CNN Decoder) | |
| 177:1 ($ls = 10,000 \times 32$) | 60.87 |
| 353:1 ($ls = 5000 \times 32$) | 60.87 |
| 1769:1 ($ls = 1000 \times 32$) | 60.71 |
| 36k:1 ($ls = 48 \times 32$) | 59.44 |

JPEG2000 achieves a significant compression ratio; however, it does not achieve more competitive results than the SSCNet model. The last model obtains remarkable results across all the metrics analyzed. From a qualitative visualization point of view, in Figure 5, we show the difference by the original source, JPEG2000, and SSCNet outputs over the first 2 bands. We highlight the higher concentration of artefacts generated by the JPEG2000 algorithm compared to that of SSCNet. The artifacts results that we noticed in the Lombardia tiles reflect across all bands by JPEG2000 image reconstructed proving the inefficiency methodology in terms of numerical analysis with data type equal to 16 bits, and, for this reason, we applied the JPEG2000 compressor for each band to have a better signal reconstruction quality (see Figure 5).

**Table 6.** Computational time (over VIRTIS-Rosetta dataset) of the SSCNet with the last convolutional layer version (SSCnet conv), the last linear layer version (SSCNet linear), and [24] GNN model. We report the number of parameters for the encoder/decoder networks (P), training time (Train T), test time encoder/decoder, and global test time (Test G time) using 1064 test instances and 128 as batch size (1000 epochs). The test of encoder/decoder time is meant as the average time of each batch.

| Model | P (M) | Train T(s) | Test T Enc(s) | Test T Dec(s) | Test G Time |
|---|---|---|---|---|---|
| SSCNet conv | 8 (enc) 6.7 (dec) | 35,020 | 0.219 | 0.125 | ∼0.34 |
| SSCNet linear | 171 (enc) 168 (dec) | 39,578 | 0.220 | 0.120 | ∼0.34 |
| [24] GNN model | 175 (enc) 0.683 (dec) | 30,074 | 0.116 | 0.090 | ∼0.20 |

SSCNet with Last Convolutional Layer

All experiments that we conducted on SSCNet have as the last part a fully connected layer in order to have the flexibility and rule-specific compression ratio, which are useful to compare our methods and results with those in the literature. In this version of the proposed method, we replace the last fully connected layer with a convolutional layer to decrease the parameters numbers of the model and avoid the square pixel effect and possible noise due to the large fully layer (see Figure 6). However, we do not have the flexibility to choose a latent space size a priori and we have to consider a generic compression ratio. In the experiment reported in Table 5, we apply a compression ratio of from 7:1 to 1769:1 over the VIRTIS-Rosetta dataset, showing the reconstruction data signals in terms of PSNR and SSIM metrics, and in Figure 7, we show a qualitative comparison with ourselves, using a compression ratio of 177:1, 27:1 and 7:1 over a specific tile (band 1). Furthermore, in Table 5, we implemented and reported the results of the autoencoder architecture based on [24] demonstrating the main improvements of the SSCNet in terms of image reconstruction. Moreover, in the SSCNet version with the last convolutional layer, we reduce the model heavily in the number of parameters than using the last linear layer, increasing its performance in terms of training time as we reported in Table 6.

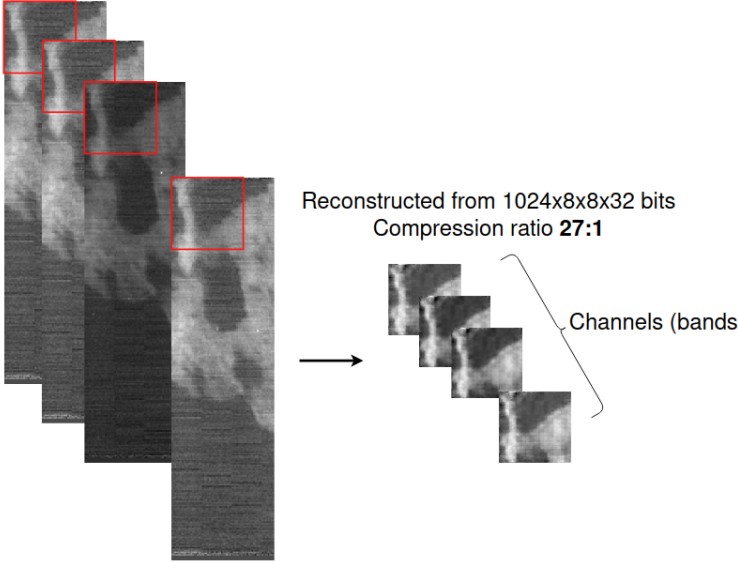

**Figure 6.** Visual comparison (VIRTIS-Rosetta dataset) using the compression ratio of 27:1 and data type 16 bits of the original source tile 432 × 64 × 64 × 32 bits (only bands 1, 2, 3, and 432 are shown) with the relative hyperspectral cube reconstructed by SSCNet from 1024 × 8 × 8 × 32 data information, using an initial preprocessing for the band normalization (max/min) to obtain a fast learning convergence.

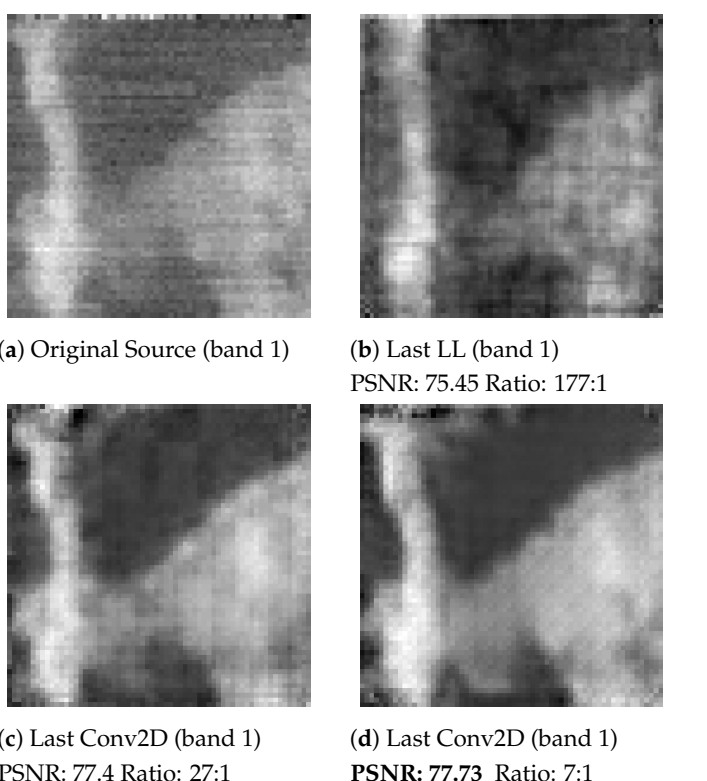

(**a**) Original Source (band 1)

(**b**) Last LL (band 1)
PSNR: 75.45 Ratio: 177:1

(**c**) Last Conv2D (band 1)
PSNR: 77.4 Ratio: 27:1

(**d**) Last Conv2D (band 1)
**PSNR: 77.73** Ratio: 7:1

**Figure 7.** Comparison using a compression ratio 177:1 (**b**) and 27:1 (**c**) and data type 32 bits (over VIRTIS-Rosetta dataset) of the original source tile (**a**). In (**b**), we show the image reconstructed (band 1) using SSCNet with the last linear layer applied and in (**c**) the SSCNet using the last 2D convolution layer. In (**d**), we use SSCNet with a compression ratio 7:1 showing improvements in terms of PSNR than the others experiments.

Finally, we show in Figure 8 the difference comparison between the spectral signal source (blue line) and the spectral signal reconstructed (red line) of two selected pixels from source and decoded information. We underline the high degree of similarity between the two plots and demonstrate the validity of the quantitative results (Table 5) obtained in all the experiments carried out. In the left image showed (only 1, 2, and 3 bands composition for visualization purpose), we detect noise, probably due to the acquisition source by the VIRTIS instrument. In conclusion, SSCNet is demonstrated to approximate the identity function also in the context of hyperspectral sources.

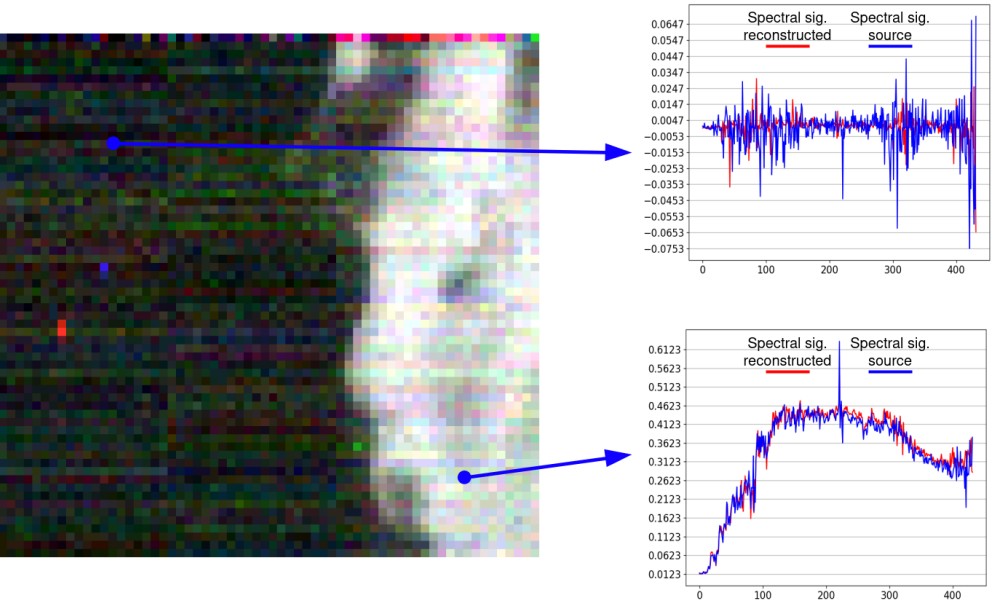

**Figure 8.** Differences between the source of the spectral signal (blue line) with the spectral signal reconstructed by SSCNet (red line) (last conv2d version) given two selected pixel coordinates (x, y) of the hyperspectral cube (source and reconstructed, respectively) on all available bands of the VIRTIS-Rosetta data (Compression Ratio 27:1).

## 4. Conclusions

Image compression covers a key role in space science and satellite imagery and, leveraging by the lack of investigation of the autoencoder models for data compression over spectral signals and various data type (greater than 8 bits), in this paper, we develop a spectral signals compressor based on the deep convolutional autoencoder (SSCNet), analyzing the learning process and evaluating it in terms of compression and spectral signal reconstruction over spectral datasets and Imagenet-ILSVRC2012 benchmark. We detected a light improvement using PReLU activation function instead of ReLU due to the learned parameters considered; furthermore, we demonstrated that SSCNet version with the last convolutional layer achieved better performance than SSCNet version with the last linear layer, even if we drastically reduced the number of parameters. We built two datasets from the ESA repository (Lombardia Sentinel-2 satellite imagery and VIRTIS-Rosetta hyperspectral signals information) and developed a Python parser useful to read and handle the calibrated data images. Furthermore, we release the PyTorch code for SSCNet, the pretrained models, and the parser software available in [28]. Extensive experiments on several benchmark datasets demonstrated the effectiveness and usability of the proposed model across RGB, multispectral, and hyperspectral sources, reporting a high compression ratio achieved and, at the same time, a great reconstruction from the compressed information. The proposed model outperformed JPEG, JPEG2000, and the related state-of-the-art techniques used as benchmarks.

**Author Contributions:** Conceptualization, R.L.G.; methodology, R.L.G.; software, R.L.G.; validation, I.G.; investigation, R.L.G.; writing—review and editing, G.C., R.L.G. and C.R.; supervision, G.C., C.R. and I.G. All authors have read and agreed to the published version of the manuscript.

**Funding:** This research received no external funding.

**Conflicts of Interest:** The authors declare no conflict of interest.

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
