# Peer review of "Hyperspectral Data Compression Using Fully Convolutional Autoencoder"

_remotesensing, doi:10.3390/rs14102472_

Round 1

Reviewer 1 Report

Dear Editor & Authors,       My comments are as follows.      In Table3, the results of the proposed SSCNet are compared with that of reference [24]. And in Table 4, only the computational time and the number of parameters of the proposed SSCNet are listed. Could you list the computational time and the number of parameters of reference [24]?The effectiveness of the SSCNet should be proved by comparion, rather than the results of itself.   Best regards,  

Author Response

Thanks for your suggestion.
We restarted the evaluation model using the pretrained model saved in the last experiment (using the [24] model) and in Tab.4 we reported the computational time (train, test enc and dec) and the number of parameters (encoder and decoder) of the reference [24].

Furthermore, we restarted each model evaluation to avoid as much as possible call function (e.g print, variable used for evaluation metric, for example, psnr computation and so on) to give you a more accurate average time in seconds.

Reviewer 2 Report

ok

Author Response

Thanks a lot.

Reviewer 3 Report

The Authors have adequately addressed the primary revision recommendations, suggestions, and comments emerged during the last review round. The technical contribution and presentation quality of the revised paper have significantly improved. Therefore, at this stage, also considered that we are already at the end of the second review round, I recommend accepting the paper as is. Congratulations!

Author Response

Thanks a lot.

Round 2

Reviewer 1 Report

The authors have answered my all concerns. I have no other comments.

This manuscript is a resubmission of an earlier submission. The following is a list of the peer review reports and author responses from that submission.

Round 1

Author Response

We appreciate and thank you for the suggestions and comments of editors and reviewers, in attached the reviews file

Yours sincerely,
Riccardo La Grassa, Ph.D.

Reviewer 2 Report

Major:
There is lack of evaluation of different Autoencoder Architectures.
It is not enough. Different configurations should be evaluated, so more results and improved discussion is expected.

Minors:
Eq.1 'log' is clear, 'log' with dot above 'l' is not.
Eq missing above l.98
Eq above l.98 please use \cdot

Table 3
0. <- 0.0000

Author Response

We appreciate and thank you for the suggestions and comments of editors and reviewers, in attach the review file.
Yours sincerely,
Riccardo La Grassa, Ph.D.

Reviewer 3 Report

  1. Image processing methods in recent years mainly use artificial intelligence, including artificial neural networks, so-called deep learning. One of the most common applications of images processing methods are satellite imaginary data volume reduce and compression. This article presents one of the problems related to Spectral Signals Compressor network based on Deep Convolutional AutoEncoder.
  2. General remarks
    1. Too many abbreviations make it difficult to follow the content of the article. Each abbreviation should be expanded the first time it appears. Not all readers need to know all abbreviations. Especially in the abstract of the article and conclusions.
    2. Please use the language of a scientific research report without personal references: “we”, “our”, and also “they” which are plenty in whole article.
    3. The titles of the drawings are too long. Descriptions of figure content and other explanations should be in the body of the article rather than in the figure title. The title should be in one sentence.
    4. The paper lacks of precise mathematical model of proposed method.
    5. However the article is well written should be carefully edited. Some remarks included below.
  3. Specific remarks
    1. There is lack of “Discussion” chapter.
    2. The final conclusions are too general and only generally summarize the research presented in the article. I suggest expanding the conclusions with more detailed findings.
    3. Please interpret the several results 0. in Table 3.
    4. The article should be carefully edited. Chapter 4 of Conclusions is interspersed several times with figures and tables. All mathematical formulas should be numbered.

Author Response

(The authors gave the same response as above.)

Round 2

Reviewer 1 Report

See the attachment.

Reviewer 2 Report

Some minors:

Eq.3 
100 <- 100%

Table.3
66.79 <- 66.790

Table.4
11.9 <- 11.90

Fig.7
Axes description missing

Reviewer 3 Report

All my remarcs are proper adressed. Atricle is ready for final processing and publication.